# Benchmarking Sequential Visual Input Reasoning and Prediction in Multimodal Large Language Models

## Abstract

Multimodal large language models (MLLMs) have shown great potential in perception and interpretation tasks, but their capabilities in predictive reasoning remain under-explored. To address this gap, we introduce a novel benchmark that assesses the predictive reasoning capabilities of MLLMs across diverse scenarios. Our benchmark targets three important domains: abstract pattern reasoning, human activity prediction, and physical interaction prediction. We further develop three evaluation methods powered by large language model to robustly quantify a model's performance in predicting and reasoning the future based on multi-visual context. Empirical experiments confirm the soundness of the proposed benchmark and evaluation methods via rigorous testing and reveal pros and cons of current popular MLLMs in the task of predictive reasoning. Lastly, our proposed benchmark provides a standardized evaluation framework for MLLMs and can facilitate the development of more advanced models that can reason and predict over complex long sequence of multimodal input.

## 1 Introduction

Multimodal Large Language Models (MLLMs) OpenAI (2023) have become a crucial area of research due to their huge potential in many applications, such as Autonomous Driving Yurtsever et al. (2020), Embodied Agents Driess et al. (2023); Wu et al. (2023), Robotics Ahn et al. (2022); Brohan et al. (2023), and many others. Current MLLMs use text-to-image pairing data to convert visual inputs into visual tokens and integrate them into existing LLMs Liu et al. (2023a). Current evaluation also focuses on the perception and reasoning ability of MLLMs given an image, such as image captioning Vinyals et al. (2016), visual question answering Antol et al. (2015), and so on. However, in many applications, MLLMs are required to reason over a sequence of visual inputs, such as the last five frames of video input to decide a robot's next action Brohan et al. (2023). This raises the question of whether or not current MLLMs, which are mostly trained on single image-text pairs, emerge the ability to reason over multiple image inputs and predict what's coming next given the context information.

Therefore, this work proposes a novel benchmark to quantify the predictive reasoning ability over three important scenarios: abstract pattern reasoning, human activity prediction, and physical interaction prediction. These three distinctive tasks require an MLLM to equip with both complex reasoning over multiple visual inputs and also use common sense world knowledge to generate highly probable outcomes. Another challenge addressed is how to robustly quantify MLLMs' predictive reasoning ability. We utilize text-only Large Language Model as a referee model and construct three paradigms of evaluation, i.e., tasks with single-golden answer, tasks with multiple-golden answers, and tasks with probabilistic outcomes.

The proposed benchmark and evaluators are tested with five popular MLLMs. Testing results confirm the soundness of the proposed tasks and evaluators. Experiment results also show surprisingly different outcomes compared to many other MLLM benchmarks, such as MME Fu et al. (2023). The authors show that simple models such as LLava Liu et al. (2023a) show the strongest generalization for predictive reasoning. They also show a huge gap between upper bound performance and

current MLLMs' ability, shining lights on the development of more advanced MLLMs in the future. In summary, this paper's contributions are:

1. We propose three challenging tasks to test the predictive reasoning of MLLMs and develop high-quality datasets for model evaluation.

2. We propose three novel evaluation methods based on large language models to quantify the predictive reasoning capabilities of MLLMs.

3. We present experiment results that validate the proposed evaluators, producing evaluation scores that highly correlate with an MLLM's predictive reasoning abilities.

4. We provide benchmark results that reveal the pros and cons of current popular MLLMs, providing insights on various MLLM network structures and potential improvement directions.

## 2 RELATED WORK

### 2.1 MULTIMODAL LARGE LANGUAGE MODELS

Multimodal Large Language Models (MLLMs) are large language models Brown et al. (2020) that can accept multimodal inputs, such as images. Notable work inlcudes LLaVA Liu et al. (2023a) that combines CLIP Radford et al. (2021) and Vicuna Chiang et al. (2023) with simple linear adapters and finetune on multimodal instruction tuning data. Later, unlike plain vision transformer patch features in LLava, MiniGPT-4 Zhu et al. (2023) employs a more sophisticated QFormer Li et al. (2023d) architecture to extract condensed visual token features as input to large language models. InstructBLIP Dai et al. (2023) further augment the training data with 13 diverse tasks including video data to improve the instruction following ability of MLLMs. mPLUG-Owl Ye et al. (2023) corrects the alignment errors and enhance its multi-turn dialogue capabilities. Conversely, Otter Li et al. (2023b) uses cross-attention architecture Awadalla et al. (2023) for more fine-grained vision-language fusion. Lastly, Lynx Zeng et al. (2023b), capable of inferring on video-level sequences, adopts prefix-tuning Li & Liang (2021) for instruction tuning Peng et al. (2023), presenting a more efficient alternative to cross-attention formats.

### 2.2 FUTURE PREDICTION BASED ON VIDEO INPUT

Video prediction is a significant area of research in computer vision. One example is the Robotic Pushing dataset Finn et al. (2016), which contains 59,000 robot interactions focused on pushing actions. This dataset enables precise video prediction by using the robot's impending actions as a conditioning factor. Another example Liang et al. (2019) studies human activity prediction based on surveillance camera videos. CLEVRER dataset Yi et al. (2020), which uses a 3D engine to simulate physical object motion and predicts collision effects based on trajectory paths. SUTD-TrafficQA Xu et al. (2021) dataset explores real-world scenarios from a first-person perspective, assessing the possibility of car accidents within the driver's field of vision. In the realm of autonomous driving, the nuScenes dataset Caesar et al. (2020) leverages urban traffic data to evaluate trajectory predictability between pedestrians and vehicles in busy city traffic environments.

### 2.2.1 MULTIMODAL LARGE LANGUAGE MODEL EVALUATION

Despite of the impressive multimodal understanding abilities of MLLMs, they also face the difficulties of evaluation due to their generative nature. VL-Checklist Zhao et al. (2022) proposes to evaluate multimodal foundation models fine-grained performance to recognize the existence of visual element given complex text prompts. Mimic-it Li et al. (2023a) introduces a 2.8 million multimodal instruction-response dataset to bolster the zero-shot performance of multimodal models. MME Fu et al. (2023) establishes a comprehensive evaluation framework focusing on perception and cognition. MMBench Liu et al. (2023b) enhances evaluation robustness by translating free-form answers into predefined choices. Finally, SeedBenchLi et al. (2023c) specializes in generative comprehension, featuring a significantly larger human-annotated multiple-choice dataset. None of prior work has focused on studying MLLM's ability to predict and reason over sequential visual input, which is the key contribution of this paper.

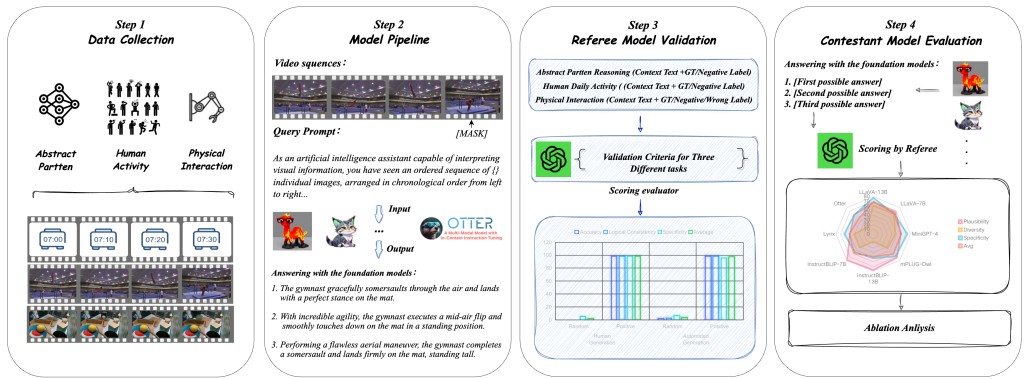

Figure 1: Overview of the proposed benchmark and method for MLLM evaluation.

# 3 PROPOSED BENCHMARK AND METHODS

We introduce three tasks, each with its own set of multiple datasets, focusing on *abstract patterns reasoning*, *human activity prediction*, and *physical interaction prediction*. The abstract patterns reasoning task is used to examine a model's raw reasoning ability; the human activity prediction task is used to measure a model's understanding of human intention and physical interaction prediction is used to test a model's grasp of physucal laws.

## 3.1 SEQUENTIAL VISUAL INPUT PREDICTION AND REASONING TASKS

### 3.1.1 TASK 1: ABSTRACT PATTERN REASONING

**Definition** In the abstract pattern reasoning task, each piece of data contains a specific pattern. Consider the following scenario: you have four images, each containing a different number of apples. The first has one, the second two, the third three, and the fourth four. The discernible pattern here is that with each subsequent image, the quantity of apples increases by one. Given this pattern, it would be reasonable to predict that the next image will contain five apples.

**Challenges** The accurate extraction of patterns from visual context data serves as a crucial prerequisite for the model's subsequent predictive inference. This is also the focus of this task. It requires the models meticulously observe the details of each image and integrate the sequential information across multiple images to find the difference and relation. In addition, our data are all created based on icons, which is also a challenge for the MLLMs.

**Data Construction** To guarantee a diverse range of patterns represented within the data, we implemented a variety of modifications and expansions across four distinct dimensions: spatial movement, rotation, quantity, and properties. For spatial movement, we utilized patterns such as variable and uniform motion, trajectory motion, and multi-entity movement. Within the rotation dimension, we integrated the fundamental concept of rotation into over ten diverse variants. For the quantity dimension, we aimed to replicate various real-world scenarios to generalize the law of quantity changes. Regarding property patterns, we developed variants from numerous perspectives, including color, numerical changes, shape, and size. Simultaneously, to ensure a diverse range of scenes within the images, we extracted various categories of entities from the icon645 dataset and incorporated them into our dataset construction process. In conclusion, we manually created 100 high-quality, pattern reasoning data entries, and using automated scripts, we generated an additional 1k pattern reasoning data entries, with each entry containing between 3-5 images.

### 3.1.2 TASK 2: HUMAN ACTIVITY PREDICTION

**Definition** In the human-centric activity task, each data point captures a snapshot of real-world human activity. The model is charged with employing common-sense reasoning to predict ensuing human actions or activities.

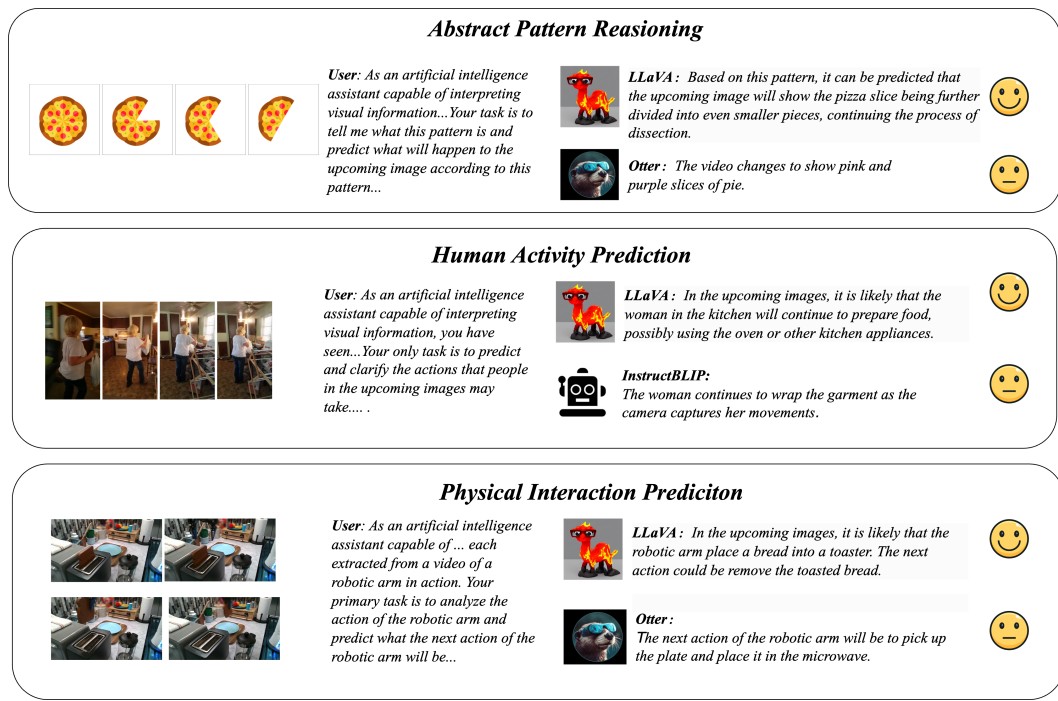

Figure 2: Visual examples of the three proposed task. The figure also shows preferred and suboptimal response from MLLMs.

**Challenges** the environmental or social context in which an activity occurs exerts a significant influence on subsequent actions, requiring models to be proficient in integrating such contextual variables. Lastly, the complexity of human behavior, guided as it is by unwritten norms and cultural codes, poses a nontrivial challenge in incorporating common-sense reasoning into the model.

**Data Construction** We utilize two key datasets: ActivityNet Captions for general daily activities and Charades for specific indoor behaviors. For ActivityNet, we standardize 309 video segments using the CLIP-L/14 model and cosine similarity for visual-text matching. Charades presents challenges due to complex annotations and overlapping timeframes. We resolve this by selecting the last verb-noun pair in the annotation for prediction. A 260-video subset of Charades underwent rigorous evaluation by domain experts to ensure dataset quality and consistency.

### 3.1.3 TASK 3: PHYSICAL INTERACTION PREDICTION

**Definition** The task at hand aims to evaluate multimodal large language models on their ability to understand and predict physical interactions and motion trajectories using the CLEVRER and RoboSet(MT-ACT) datasets. For CLEVRER, models must decipher 5-second sequences capturing object motions and predict potential outcomes, such as collisions or changes in trajectory. In the context of RoboSet(MT-ACT), models are tasked with predicting the subsequent actions of a robotic arm based on an ordered series of subtasks in a kitchen environment.

**Challenges** In RoboSet(MT-ACT), the model must identify the current robotic arm activity and predict its next immediate step. For CLEVRER, accurate interaction prediction relies on discerning unique object attributes and 3D spatial awareness. Our frame-sampling method further necessitates inferring latent variables like velocity from sparse cues. Thus, effective performance requires expertise in object identification, spatial cognition, and temporal reasoning. In the CLEVRER dataset, unique object attributes necessitate precise recognition for accurate interaction prediction. The model must also demonstrate nuanced 3D spatial awareness to interpret object positions and movements within complex scenes. Furthermore, our frame-sampling approach requires the model to infer latent variables such as velocity and acceleration from sparse visual cues. Therefore, effec-

tive task performance mandates concurrent expertise in object identification, spatial cognition, and temporal reasoning for reliable short-term physical interaction prediction.

**Data Construction** Regarding Roboset(MT-ACT), we eliminated incoherent datas, resulting in 13 logically consistent subtask combinations from seven original activities. Each combination includes 25 or 50 variants, and we perform precise frame positioning manually dissect actions into 3 to 5 discrete frames per subtask combination. For evaluation, this led to a test set of 301 samples and a training set of 756 samples.

The CLEVRER dataset features 5-second videos of object motion with unique attributes. We employ a frame-sampling strategy for evaluating multimodal language models in short-term prediction tasks. We chose 300 for focused assessment.

## 3.2 PREDICTIVE REASONING EVALUATION VIA LARGE LANGUAGE MODELS

Prior to deploying the large language model as an evaluator $g$, its effectiveness is rigorously quantified across three specific tasks—Abstract Pattern Reasoning(A), Human Activity(H), and Physical Interaction(P). For this purpose, we define $y$ as a discrete variable, selected from the task space $\mathcal{Y}$, i.e., $y \in \mathcal{Y} = \{A, H, P\}$. Equations 1 to 3 mainly quantify the effectiveness of the evaluator for these tasks.

We introduce a multimodal large language model $f(\cdot)$, which takes a set of visual features $F$ from the feature space $\mathcal{F}$ and text prompts $Q$ from the query space $\mathcal{Q}$, both tailored for the three types of tasks $(y)$:

$$F_{i,d} = \phi(I_{i,d}), \quad F_{i,d} \in \mathcal{F}, \quad \forall i \in \text{Segments of video } d, \quad \forall d \in \mathcal{D}_{\text{val}} \tag{1}$$

In this context, $D_d$ and $T_d$ represent the textual descriptions corresponding to each segment and the final segment of video $d$, respectively. Together, they form the complete description of the video. The evaluator $g$ serves to quantify the performance of the multimodal model $f$. For each task $y$, we use the target text $T_{y,d}$ and the descriptive text $D_{y,d}$ for each data entry $d$ in the validation dataset $\mathcal{D}_{\text{val}}$ as inputs to generate a score:

$$S_{y,\text{val},d} = g(T_{y,d}, D_{y,d}, y, o) \tag{2}$$

This notation communicates that $o$ is optional and may or may not be provided in the function call. The effectiveness of the evaluator is quantified by calculating the average score over $\mathcal{D}_{\text{val},y}$:

$$\text{Val}_y(S_{\text{val}}) = \frac{1}{|\mathcal{D}_{\text{val},y}|} \sum_{d \in \mathcal{D}_{\text{val},y}} S_{y,\text{val},d} \tag{3}$$

We define $\mathcal{H}$ as the function space containing all possible contestant models, and we assume the existence of an optimal model $f^*$. Equations 4 to 7 primarily use the evaluator to quantify scores for the dataset across the tasks.

Each $f \in \mathcal{H}$ operates in the visual feature space $\mathcal{F}$ and the query space $\mathcal{Q}$, and outputs to the textual prediction space $\mathcal{T}$:

$$\hat{T}_{y,d} = f(\{F_{i,d}\}, Q_{y,d}, y), \quad f \in \mathcal{H} \tag{4}$$

Here, $P(\hat{T}_{y,d}|\{F_{i,d}\}, Q_{y,d})$ represents the probability of the predicted label $\hat{T}_{y,d}$ given a set of visual features $\{F_{i,d}\}$ and a text prompt $Q_{y,d}$. The referee model $g : \mathcal{T} \times \mathcal{D} \times \mathcal{Y} \rightarrow \mathcal{S}$ operates in the textual prediction space $\mathcal{T}_y$. Finally, the evaluator $g$ operates in the textual prediction space $\mathcal{T}_y$, the description space $\mathcal{D}_y$, and the score space $\mathcal{S}_y$:

$$S_{y,d} = g(D_{y,d}, \hat{T}_{y,d}, y, o) \tag{5}$$

In the given equations, $P(S_{y,d}|D_{y,d}, \hat{T}_{y,d}, y)$ serves as the conditional probability of the score $S_{y,d}$ given the description $D_{y,d}$, predicted label $\hat{T}_{y,d}$, and the type $y$. This metric provides a quantifiable measure of how well the model's prediction aligns with the given description and type, thus enabling a more nuanced evaluation. Like $S_{y,d}$, it is computed for each type $y \in \mathcal{Y}$, making it a flexible evaluation metric that can be tailored to different types of data and tasks.

$$\text{Avg}(S_y) = \frac{1}{|\mathcal{D}_y|} \sum_{d \in \mathcal{D}_y} S_{y,d} \quad \text{where } y \in \{A, H, P\} \tag{6}$$

**Single Gold Answer Evaluator (SGAE) :** is specifically designed for assessing model response against deterministic ground truths, primarily in abstract pattern reasoning task. Given that there are established Ground Truth labels for this task, it's essential for the evaluator to incorporate these Ground Truth labels during scoring, in addition to the context information $D_y$.

The expectation is for the model to distill the pattern from visual data and make predictive inferences accordingly. Therefore, we should consider these two aspects separately. For pattern extraction, we adopt **Logical Consistency** as the evaluation metric, which gauges whether the model's response aligns logically with the ground truth pattern. To assess predictive performance, we use **Accuracy** as a metric. Given that each image sequence in pattern reasoning data contains a clear pattern with little redundant information and the content of the next image to be predicted is also specific with no multiple possible scenarios, the accuracy concept fits our context aptly. Furthermore, considering that model responses might differ in their level of detail, we use **Specificity** as a measure of the model's response granularity.

**Probabilistic Prediction Evaluator (PPE):** is optimized for assessing multimodal language models in the realm of human activities. It relies chiefly on two metrics: **Plausibility** and **Diversity**, supplemented by **Specificity**. Plausibility assesses the model's aptitude for generating contextually plausible human actions. Diversity gauges the model's ability to generate a spectrum of such plausible outcomes, capturing the multi-faceted nature of human behavior. These metrics work in tandem to provide a balanced evaluation, enhancing model generalization and mitigating overfitting risks. The supplementary Specificity metric adds granularity to this assessment.

**Multiple Gold Answer Evaluator (MGAE):** serves as an amalgamation of the first two types, blending factual rigor with predictive utility. Although physical interaction data inherently offers objective facts, we contend that model efficacy should not be confined to this narrow scope. For instance, if a model successfully predicts a collision occurring two seconds into the future, it should be additionally rewarded for accurate extrapolations thereafter. In this paradigm, we employ accuracy (ACC) as a key metric, but diverge in our approach by adopting a point-based scoring system. Full accuracy scores are granted if the model's range of possible outcomes includes the ground truth. Furthermore, any generated results that adhere to the logical constraints of physical laws are accorded high scores for logical consistency. Therefore, regarding this evaluator, on one hand, the model's output encompasses a range of possibilities. On the other hand, the Ground Truth is additionally incorporated as a reference for evaluation.

## 4 EXPERIMENTS

In this section, we conducted a comprehensive evaluation of five MLLMs (LLaVA Liu et al. (2023a), MiniGPT-4 Zhu et al. (2023), mPLUG-Owl Ye et al. (2023), InstructBLIP Dai et al. (2023), Lynx Zeng et al. (2023a), Otter Li et al. (2023b)) on our three tasks.

For model settings, such as temperature, we used the default parameters provided in the demo code of each model. Considering the differences between the models, we customized two sets of queries for each dataset, corresponding to the situations when the image sequence is input as images and as a video. Given that some models' default system prompts don't support multiple images, we modified them for compatibility. The final versions of the system prompts we used can refer to appendix.

The results of our experiments reveal: 1) From the model perspective, LLaVA significantly outperformed the other models on our datasets. 2) From the task perspective, the MLLMs excelled at tasks like human daily activities, but performed poorly on abstract, non-realistic datasets.

**All models struggle to grasp abstract pattern reasoning.** The results of the abstract pattern reasoning dataset which are presented in Table 1 are generally quite low. The Specificity dimension, which evaluates the level of detail in the model's responses, exhibits somewhat higher scores. However, the results are notably subpar in areas such as Logical Consistency, which involves multi-image pattern extraction, and prediction accuracy. Given that the MLLMs we've examined have not undergone training in multi-image scenarios, their strength primarily lies in understanding image content. Consequently, the disparity in these scores is expected.

**Models achieve stronger performance human activity prediction.** This is likely attributable to the semantic richness and inherent interpretability of human activities. These characteristics facilitate

| Models | Human Generated | | | | Automated Generated | | | |
|---|---|---|---|---|---|---|---|---|
| | Acc. | Logic. | Spec. | Avg. | Acc. | Logic. | Spec. | Avg. |
| LLaVA-13B | **10.60** | **18.80** | **25.0** | **18.13** | **4.14** | **10.64** | **18.08** | **10.95** |
| LLaVA-7B | 4.00 | 11.60 | 17.20 | 10.93 | 2.14 | 7.68 | 13.10 | 7.64 |
| MiniGPT-4 | 3.80 | 6.40 | 11.20 | 7.13 | 2.10 | 2.80 | 7.42 | 4.11 |
| mPLUG-Owl | 8.60 | 16.40 | 21.20 | 15.40 | 1.68 | 6.20 | 13.06 | 6.98 |
| InstructBLIP-13B | 8.80 | 17.60 | 17.20 | 14.53 | 3.14 | 9.96 | 10.56 | 7.89 |
| InstructBLIP-7B | 4.20 | 8.40 | 8.60 | 7.40 | 1.08 | 5.26 | 5.56 | 3.97 |
| Lynx | 2.60 | 10.00 | 13.80 | 8.80 | 0.74 | 3.00 | 6.74 | 3.49 |
| Otter | 0.80 | 2.20 | 3.40 | 2.13 | 0.16 | 1.20 | 1.50 | 0.95 |

Table 1: Results of **abstract pattern reasoning task**. "Acc, Logic, Spec, Avg" refers to Accuracy, Logical Consistency, Specificity and Average score respectively.

more effective salient information capture and enable greater imaginative reasoning. The open-ended evaluation criterion implemented for this task, which is devoid of a standardized answer key, addresses the wide range of potential outcomes in human activities. This approach also elevates the probability of evaluators awarding higher scores.

| Models | ActivityNet Captions | | | | Charades | | | |
|---|---|---|---|---|---|---|---|---|
| | Plausibility | Diversity | Specificity | Avg. | Plausibility | Diversity | Specificity | Avg. |
| LLaVA-13B | 68.16 | 61.42 | **77.73** | **69.1** | 59.23 | **58.08** | **76.27** | **64.53** |
| LLaVA-7B | 67.51 | **62.14** | 62.14 | 63.98 | **62.35** | 36.38 | 65.42 | 54.67 |
| MiniGPT-4 | 58.96 | 47.54 | 73.43 | 59.98 | 37.38 | 24.69 | 60.85 | 40.95 |
| mPLUG-Owl | 71.91 | 43.2 | 67.73 | 60.95 | 59.19 | 26.81 | 71.08 | 52.36 |
| InstructBLIP-13B | 77.48 | 16.44 | 52.1 | 48.67 | 58.84 | 3.36 | 52.64 | 39.28 |
| InstructBLIP-7B | **79.55** | 41.17 | 55.47 | 58.73 | 58.62 | 4.85 | 51.85 | 38.44 |
| Lynx | 50.29 | 41.04 | 62.59 | 51.31 | 39.12 | 23.77 | 54.00 | 38.96 |
| Otter | 57.99 | 19.61 | 57.99 | 45.2 | 54.54 | 9.15 | 57.62 | 40.44 |
| Vicuna-13B | 90.91 | 68.67 | 75.92 | 78.5 | - | - | - | - |
| Vicuna-7B | 89.19 | 64.3 | 73.07 | 75.52 | - | - | - | - |

Table 2: Results of **human activity prediction task**. We evaluate the predictive inference capabilities of Vicuna which is a unimodal language model on the ActivityNet Captions dataset for comparison.

**Models perform poorly in physical interaction prediction.** In the physical interactions task, models generally exhibit low accuracy scores. The complexity of the CLEVRER dataset, necessitating the inference of potential velocity variables and the differentiation of object attributes such as shape and color, poses significant challenges that most models find difficult to surmount effectively. The score disparity between the two dimensions on the Roboset dataset is quite pronounced. As shown in 3, most models display high Logical Consistency, indicating their basic understanding of the scenario under the robotic arm, but they have extremely low accuracy rates. Upon examining the models' outputs, we found that they tend to focus more on conjecturing possible actions based on the current scenario, rather than further reasoning the robotic arm's movements according to the visual context.

**Image MLLMs outperform Video MLLMs** From the results of our evaluations, models that had only been trained on image data, such as LLaVA Liu et al. (2023a) and mPLUG-Owl Ye et al. (2023), outperformed those trained with video data, like OtterLi et al. (2023b), which surprisingly underperformed. This gap is particularly noticeable in Table 1 and Table 3. Similar conclusions have also been demonstrated in other works Li et al. (2023c;e); Maaz et al. (2023).

**LLaVA demonstrates exceptional performance across all tasks** LLaVA Liu et al. (2023a) directly employs tokens from CLIP Radford et al. (2021), combined with language tokens through a simple

| Models | CLEVRER | | | | RoboSet (MT-ACT) | | |
|---|---|---|---|---|---|---|---|
| | Acc. | Logic. | Spec. | Avg. | Acc. | Logic. | Avg. |
| LLaVA-13B | 10.49 | 34.46 | 51.84 | 32.26 | 3.65 | 53.82 | 28.73 |
| LLaVA-7B | **25.56** | **38.50** | 44.29 | **36.12** | **10.96** | 57.14 | 34.05 |
| MiniGPT-4 | 0.00 | 0.55 | 10.52 | 3.69 | 3.99 | 44.58 | 24.29 |
| mPLUG-Owl | 1.18 | 25.65 | **52.47** | 26.43 | 3.65 | 50.96 | 27.3 |
| InstructBLIP-13B | 19.26 | 33.89 | 21.73 | 24.96 | 7.64 | **65.45** | **36.55** |
| InstructBLIP-7B | 18.22 | 27.05 | 20.31 | 21.86 | 6.81 | 41.99 | 24.40 |
| Lynx | 6.27 | 20.96 | 20.89 | 16.04 | 7.64 | 49.24 | 28.44 |
| Otter | 10.34 | 25.86 | 26.90 | 21.03 | 3.99 | 52.36 | 28.18 |

Table 3: Results of **physical interaction prediction task**. We show results on both CLEVER and RoboSet datasets.

linear mapping. The adoption of these high-quality, pre-trained tokens could be a key factor in the model's superior generalization performance, outperforming both the QFormer and Cross-Attention architectures."Additional factors such as leveraging GPT-4 generated instrument for complex reasoning, end-to-end training for holistic optimization, and multi-task fine-tuning further augment its adaptability and efficacy across various scenarios.

**Limitation of current popular model architecture** MiniGPT-4 Zhu et al. (2023) exhibits a higher propensity for repetitive text generation compared to other models, with limited effective length of its textual output. Consequently, we imposed a token limit of 150 for MiniGPT-4 Zhu et al. (2023), while maintaining a 512-token constraint for other models. This limitation in MiniGPT-4 Zhu et al. (2023) may be attributed to its Vision Transformer Dosovitskiy et al. (2020) architecture being trained on lower-resolution data, rendering it less adaptive to high-resolution images. Similar conclusions have been articulated in Chen (2023).

## 5 ABLATION AND ANALYSIS

### 5.1 EFFECTIVENESS OF THE PROPOSED EVALUATORS

We designed three experiments to separately validate the effectiveness of the three evaluators we proposed. For each experiment, the original dataset sizes were significantly larger than the retained evaluation subsets for each task. Specifically, we sampled 15% of the data for the abstract pattern reasoning task, while for the other tasks (human activity prediction and physical interaction prediction), we sampled data that matched the scale of the evaluation datasets.

| Scores | Human Generated | | Automated Generated | |
|---|---|---|---|---|
| | Random | Positive | Random | Positive |
| Accuracy | 0.00 | 98.67 | 2.13 | 98.80 |
| Logical Consistency | 0.00 | 98.67 | 2.67 | 98.53 |
| Specificity | 5.33 | 98.67 | 6.67 | 96.13 |
| Average | 1.78 | 98.67 | 3.82 | 97.82 |

Table 4: Results of **single gold answer evaluator** verification. "Random" refers to the scores of randomly selected negative samples. "Positive" refers to the scoring results of approximate answers generated by GPT-4.

For Single Gold Answer Evalautor (SGAE), we initially employed GPT-4 to generate descriptions that are semantically similar to the ground truth labels, which served as positive samples for our dataset. Concurrently, we randomly sampled labels from the dataset to represent the negative samples. These positive and negative samples were then separately scored by SGAE. Our experimental results are displayed in Table 4. From the data, we observed significant variability in the scores, which substantiates the effectiveness of SGAE.

| ActivityNet Captions | Plausibility | Diversity | Specificity | Avg. |
|:---:|:---:|:---:|:---:|:---:|
| random | 23.93 | 5.60 | 61.46 | 30.33 |
| positive | 82.10 | 2.93 | 60.17 | 48.40 |

Table 5: Results of **probabilistic prediction evaluator** verification. "Random" refers to the scores of randomly selected negative samples. "Positive" refers to the true labels inherent in the video prediction segments.

For Probabilistic Prediction Evaluator (PPE), we use the caption of each event-predicting segment as a positive sample and randomly select other segments from the same dataset as negative samples. Table5 reveals a significant variance in plausibility scores, while other dimensions show minimal differences, underscoring the Evaluator's discriminative and effective assessment across dimensions.

The primary characteristic of Multiple Gold Answer Evaluator (MGAE), in comparison to other Evaluators, lies in its method of assessing accuracy. To verify the effectiveness of its scoring system for accuracy, we employ the same approach as SGAE to obtain semantically similar answers. However, when it comes to input, MGAE utilizes a combination of one semantically similar answer and three random unrelated answers as positive sample and four random unrelated samples as negative sample. For inputs that include positive samples, we aim for an accuracy output of 1, otherwise, the output should be 0. Our experimental results are shown in Table 6.

| Datasets | Positive | Negative | Total | Acc. |
|:---:|:---:|:---:|:---:|:---:|
| CLEVRER | 97 | 64 | 427 | 90.5 |
| RoboSet(MT-ACT) | 19 | 5 | 158 | 96.2 |

Table 6: Results of **multiple gold answer evaluator** verification. The term "Positive" refers to the number of instances in which the evaluator incorrectly classified positive samples. Conversely, "Negative" represents the number of instances where the evaluator mistakenly deemed negative samples as correct. In regards to the two datasets related to physical interactions, samples were taken from each at a rate of 15%.

## 5.2 VISUAL INPUT VS. TEXT-ONLY INPUT

MLLMs are fine-tuned based on the foundational unimodal Large Language Models (LLMs). Therefore, the predictive reasoning capability of the foundational LLMs greatly impacts MLLMs. We chose to experiment with the base LLM of the LLaVA model which is Vicuna v1.1 proposed in Chiang et al. (2023). Regarding input, we directly fed the text description of the visual context into Vicuna, and used a similar set of queries to guide Vicuna in making predictive inferences. Table 2 presents the experimental results of Vicuna. Most notably, its score in the Plausibility dimension significantly surpassed that of MLLMs. Thus, the understanding of visual information remains one of the bottlenecks in the performance of MLLMs.

## 6 CONCLUSION

In this study, we establish a robust benchmarking framework consisting of three tasks to rigorously evaluate the predictive reasoning abilities of Multimodal Large Language Models (MLLMs). To ensure precise quantification, we introduce and empirically validate three novel evaluators. Our results diverge significantly from existing benchmarks, providing new insights into the predictive reasoning capabilities and limitations of MLLMs. Importantly, we identify a substantial gap between current MLLM performance and theoretical upper bounds, underlining the need for future advancements. This work sets a new benchmark for MLLM assessment and provides a foundation for targeted research in this rapidly evolving domain.

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
