# 1  SYSTEM PROMPTS FOR MLLMS

**LLaVA**

A chat between a curious human and an artificial intelligence assistant. The assistant gives helpful, detailed, and polite answers to the human's questions.

- - - - - - - - - - - - - - - - - - - - - - - - - - - - - - - - - - - - - - - - - - - - - - -

USER: $\langle im\_start \rangle \langle image \rangle \langle im\_end \rangle \langle im\_start \rangle \langle image \rangle \langle im\_end \rangle \langle query \rangle$
ASSISTANT:

---

**MiniGPT-4**

Give the following images: $\langle /Img \rangle \langle ImageContent \rangle \langle /Img \rangle$
You will be able to see the images once I provide it to you. Please answer my questions.

- - - - - - - - - - - - - - - - - - - - - - - - - - - - - - - - - - - - - - - - - - - - - - -

Human: $\langle /Img \rangle \langle ImageHere \rangle \langle /Img \rangle \langle /Img \rangle \langle ImageHere \rangle \langle /Img \rangle \langle query \rangle$
ASSISTANT:

---

**mPLUG-Owl**

The following is a conversation between a curious human and AI assistant. The assistant gives helpful, detailed, and polite answers to the user's questions.

- - - - - - - - - - - - - - - - - - - - - - - - - - - - - - - - - - - - - - - - - - - - - - -

Human: $\langle image \rangle \langle image \rangle$
AI:

---

**InstructBLIP**

$\langle image \rangle \langle image \rangle \langle query \rangle$

---

**Lynx**

$\langle image \rangle \langle image \rangle$
User: $\langle query \rangle$
Bot:

---

**Otter**

$\langle image \rangle$
User: $\langle query \rangle$
GPT:

### Pattern Reasoning

**Image Quary** As an artificial intelligence assistant capable of interpreting visual information, you have seen an ordered sequence of individual images, arranged in chronological order from left to right. There is a pattern in these images. Your task is to tell me what this pattern is and predict what will happen to the upcoming image according to this pattern. Please provide a clear sentence describing this pattern and your prediction without requiring any additional information or factual reporting.

**Video Quary** As an artificial intelligence assistant capable of interpreting visual information, you have seen an ordered sequence of individual images, arranged in chronological order from left to right. There is a pattern in these images. Your task is to tell me what this pattern is and predict what will happen to the upcoming image according to this pattern. Please provide a clear sentence describing this pattern and your prediction without requiring any additional information or factual reporting.

### Human Centric - ActivityNet

**Image Quary** As an artificial intelligence assistant capable of interpreting visual information, you have seen an ordered sequence of individual images, arranged in chronological order from left to right, which are an ordered sequence of individual images from the same video clip. Your task is to predict and clarify the actions that people in the upcoming images may take. You can give multiple possible results. For each answer, please provide a clear sentence describing the expected action without requiring any additional information or factual reporting. Your responses should follow an enumerated format that emphasizes the possibility of multiple answers, for example: '1. [First possible answer]', '2. [Second possible answer]', '3. [Third possible answer]', and so on.

**Video Quary** As an artificial intelligence assistant capable of interpreting visual information, you have seen a video. Your task is to predict and clarify the next actions that people in the video may take. You can give multiple possible results. For each answer, please provide a clear sentence describing the expected action without requiring any additional information or factual reporting.

## Human Centric - Charades

**Image Quary** As an artificial intelligence assistant capable of interpreting visual information, you have seen an ordered sequence of individual images, arranged in chronological order from left to right, which are an ordered sequence of individual images from the same video clip. Your only task is to predict and clarify the actions that people in the upcoming images may take. You can give multiple possible results. For each answer, please provide a clear sentence describing the expected action without requiring any additional information or factual reporting. Use serial numbers to differentiate between your answers, like this:1.[].2.[].

**Video Quary** As an artificial intelligence assistant capable of interpreting visual information, you have seen a portion of a video clip. Your only task is to predict and clarify the actions that people in the upcoming segments of the video may take. You can give multiple possible results. For each answer, please provide a clear sentence describing the expected action without requiring any additional information or factual reporting.

## Physical Interaction - CLEVRER

**Image Quary** As an artificial intelligence assistant capable of interpreting visual information, you have seen a sequence of five chronologically arranged images, each extracted from the concluding frame of each second in a 5-second video. Your primary task is to analyze the motion trajectories depicted in these images, and predict which targets are on a collision course and which will continue unscathed. You can give multiple possible results. For each answer, please provide a clear sentence describing the expected action without requiring any additional information or factual reporting. Your responses should follow an enumerated format that emphasizes the possibility of multiple answers, for example: '1. [First possible answer]', '2. [Second possible answer]', '3. [Third possible answer]', and so on.

**Video Quary** As an artificial intelligence assistant capable of interpreting visual information, you have seen a video of physics-driven interactions among various objects. Your primary task is to analyze the motion trajectories depicted in these video, and predict which targets are on a collision course and which will continue unscathed. You can give multiple possible results. For each answer, please provide a clear sentence describing the expected action without requiring any additional information or factual reporting.

## Physical Interaction - RoboSet

**Image Quary** As an artificial intelligence assistant capable of interpreting visual information, you have seen a sequence of five chronologically arranged images, each extracted from the concluding frame of each second in a 5-second video. Your primary task is to analyze the motion trajectories depicted in these images, and predict which targets are on a collision course and which will continue unscathed. You can give multiple possible results. For each answer, please provide a clear sentence describing the expected action without requiring any additional information or factual reporting. Your responses should follow an enumerated format that emphasizes the possibility of multiple answers, for example: '1. [First possible answer]', '2. [Second possible answer]', '3. [Third possible answer]', and so on.

**Video Quary** As an artificial intelligence assistant capable of interpreting visual information, you have seen a video of physics-driven interactions among various objects. Analyze the motion trajectories of the targets in the video and predict the instances of collision and collision avoidance. You can give multiple possible results. For each answer, please provide a clear sentence describing the expected action without requiring any additional information or factual reporting.

## 3 EVALUATOR PROMPTS

**Abstract Pattern Reasoning**

Apply the following three criteria to assess the response ($\langle A \rangle$) of a multimodal model based on a provided pattern ($\langle P \rangle$):
Here's the format for the information you'll receive:
$\langle P \rangle$: Description of the pattern
$\langle A \rangle$: Response from the model
1. LogicalConsistency: Determine whether $\langle A \rangle$ gives a precise description of the pattern in $\langle P \rangle$. Rate it from 0 (completely inconsistent or irrelevant) to 5 (completely match with $\langle P \rangle$).
2. Accuracy: Assess if $\langle A \rangle$ accurately predicts the content of the next image based on $\langle P \rangle$'s guidelines. Rate from 0 (utterly inaccurate or irrelevant) to 5 (complete match with $\langle P \rangle$).
3. Specificity: Evaluate the level of detail in $\langle A \rangle$. It should provide a clear and concise pattern and future image, avoiding ambiguity. Score from 0 (utterly irrelevant or inaccurate to $\langle P \rangle$) to 5 (highly detailed and consistent with $\langle P \rangle$).
Note: The model has been told that the images are in chronological order and the next image will follow the same pattern. Therefore, ignore any descriptions in $\langle A \rangle$ related to the above information when scoring the model's responses. For example, if the model's response is "The pattern is a series of images that are arranged in chronological order from left to right. The next image will still follow this pattern.". All scores should be 0 points. Because it only confirms prior information.
For each category, record your scores and reasons as follows:
1. LogicalConsistency: score: []. reason: [].
2. Accuracy: score: []. reason: [].
3. Specificity: score: []. reason: [].

- - - - - - - - - - - - - - - - - - - - - - - - - - - - - - - - - - - - - - - - - - - -

**ActivityNet**

Hello GPT,

As an experienced evaluator, we would like you to score the prediction of a multimodal model. This model aims to predict what the person in the images will do next, after considering the first few images.
Please find the descriptions of the images below:
{context image captions}
The model predicted the next action as: {model answer}
We would like you to evaluate this prediction based on the following criteria:
1.Plausibility: This dimension evaluates whether the model's predicted content is realistic, logical, and coherent with the preceding images' content. A prediction can be highly specific, but if it doesn't align coherently or logically with the prior images, its plausibility is low. For instance, if previous images described a man walking by the sea, and the model predicts the next image to suddenly place him on Mars, this would clearly be implausible.
Please rate it on a scale of 0 to 5, where 0 means 'the prediction is entirely implausible, showing no logical connection to the preceding images or introducing wildly unrealistic elements' And 5 means 'the prediction seamlessly aligns with the preceding images, showing a clear and realistic continuation or development from the previous context'.
2.Specificity: This dimension focuses on the level of detail in the model's predictions. Note that the plausibility of predicting content should not be considered in this criterion.This dimension should be scored independently of the first dimension.You do not need to consider whether these details are relevant to the context provided by the images.Predictions with high specificity aren't just vague or general but provide clear, detailed information. Using the previous example, a general prediction might be ïhe man continues to walk,̈while a more specific one might be ïhe man walks along the golden beach, shoes in hand, leaving footprints on the wet sand.̈
Please rate it on a scale of 0 to 5, where 0 means ' The prediction is entirely general, with no specific details or clarity on the scenario's continuation.' And 5 means ' The prediction provides a richly detailed and clear continuation, shedding light on specific elements, actions, or characteristics'.
3.Diversity: This evaluates whether the model can offer multiple, different, yet plausible answers for the same input. In real life, many scenarios can unfold in various ways. Thus, a good model should be able to capture this diversity and not produce the exact same answer every time.
Please rate it on a scale of 0 to 5, where 0 means ' The model always gives the same or very similar answers, showing no diversity in its predictions' And 5 means ' The model consistently offers multiple distinct and plausible continuations for similar inputs, capturing a wide range of possibilities'.
For each criterion, please also provide your rationale behind the score. Thanks!
The format of your answer is as follows:
1. Plausibility: score:[].reason:[].
2. Specificity: score:[].reason:[].
3. Diversity: score:[].reason:[].

----------------------------------------------------------------

**Charades**

Hello GPT,
As an experienced evaluator, we would like you to score the prediction of a multimodal model. This model aims to predict what the person in the images will do next, after considering the first few images.
Please find the descriptions of the images below:
captions
The model predicted the next action as: answer
We would like you to evaluate this prediction based on the following criteria:
1.Plausibility: This dimension evaluates whether the model's predicted content is realistic, logical, and coherent with the preceding images' content. A prediction can be highly specific, but if it doesn't align coherently or logically with the prior images, its plausibility

is low. For instance, if previous images described a man walking by the sea, and the model predicts the next image to suddenly place him on Mars, this would clearly be implausible.

Please rate it on a scale of 0 to 5, where 0 means 'the prediction is entirely implausible, showing no logical connection to the preceding images or introducing wildly unrealistic elements' And 5 means 'the prediction seamlessly aligns with the preceding images, showing a clear and realistic continuation or development from the previous context'.

2.Specificity: This dimension focuses on the level of detail in the model's predictions. Note that the plausibility of predicting content should not be considered in this criterion.This dimension should be scored independently of the first dimension.You do not need to consider whether these details are relevant to the context provided by the images.Predictions with high specificity aren't just vague or general but provide clear, detailed information. Using the previous example, a general prediction might be "the man continues to walk,"while a more specific one might be "the man walks along the golden beach, shoes in hand, leaving footprints on the wet sand."

Please rate it on a scale of 0 to 5, where 0 means ' The prediction is entirely general, with no specific details or clarity on the scenario's continuation.' And 5 means ' The prediction provides a richly detailed and clear continuation, shedding light on specific elements, actions, or characteristics'.

3.Diversity: This evaluates whether the model can offer multiple, different, yet plausible answers for the same input. In real life, many scenarios can unfold in various ways. Thus, a good model should be able to capture this diversity and not produce the exact same answer every time.

Please rate it on a scale of 0 to 5, where 0 means ' The model always gives the same or very similar answers, showing no diversity in its predictions' And 5 means ' The model consistently offers multiple distinct and plausible continuations for similar inputs, capturing a wide range of possibilities'.

For each criterion, please also provide your rationale behind the score. Thanks!

The format of your answer is as follows:

1. Plausibility: score:[].reason:[].
2. Specificity: score:[].reason:[].
3. Diversity: score:[].reason:[].

------------------------------------------------

**CLEVRER**

As an expert evaluator, your task is to analyze the prediction results of a multimodal model. This model predicts whether certain targets in the keyframes of a 5-second video will collide or avoid collision in the next two seconds. You will receive the model's prediction for each individual sample and the corresponding true label.

These are the evaluation dimensions:

1. Specificity Evaluate the level of detail regarding the collision information in ¡prediction¿. The prediction should provide a clear and unambiguous description, specifically concerning the details of the collision, such as when, where, or what objects are involved.

Score from 0 (no specific or irrelevant collision information provided) to 5 (comprehensive and specific collision information provided, including involved objects, time, and location).

2. Logical Consistency Evaluate whether the prediction is logically consistent and based on the object attributes (such as shape, coordinates, movement speed) presented in the fifth picture.

Score from 0 (inconsistent or irrelevant to the attributes in the fifth picture) to 5 (completely consistent and based on the attributes in the fifth picture).

3. Accuracy Note that the prediction results can be diverse, indicating that the model has made predictions about various possible scenarios for each individual sample. However, the true label for each sample is unique. Evaluate whether there is an answer in the prediction that is semantically consistent with the true label of each sample.

If there is an answer in the prediction that is semantically consistent with the true label, then the prediction for that sample scores a 1 (score_i=1). If there is no answer in the prediction

that is semantically consistent with the true label, then the prediction for that sample scores a 0 (score_i=0). Your primary task is to provide these scores without delving too deep into the evaluation details.
Required Information for Evaluation:
Prediction: {sentence_text}
True Label: {ground_truth_text}
Attributes: {object_descriptions}

Your score for each dimension should be a single number, not a list. Please evaluate the prediction for each sample and return your scores in this format:
Specificity: score: [].
Logical Consistency: score: [].
Accuracy: score: [].
Your score for each dimension should be a single number, not a list

- - - - - - - - - - - - - - - - - - - - - - - - - - - - - - - - - - - - - - - - - - - - - - - - - - - - - - - -

**RoboSet**

As an expert evaluator, your task is to analyze the prediction results of a multimodal model. This model predicts the next movement of the robotic arm based on frames extracted from a video of the robotic arm's movements. You will receive the model's prediction for each individual sample and the corresponding true label.
These are the evaluation dimensions:
1. Logical Consistency
Evaluate whether the prediction is based on a complete understanding of the details of each previous robotic arm action. These details include: the specific category of objects the robotic arm is grabbing and the objects to which the robotic arm is moving. When scoring, you should focus on the specific behavior of the robotic arm and not on further associative reasoning.
Score from 0 (There is no understanding of the previous actions of the robotic arm or the predictions given are not related to the previous actions of the robotic arm.) to 5 (completely understand the details of each previous action of the robotic arm.).
2. Accuracy
Note that the prediction results can be diverse, indicating that the model has made predictions about various possible scenarios for each individual sample. However, the true label for each sample is unique. Evaluate whether there is an answer in the prediction that is semantically consistent with the true label of each sample.
If there is an answer in the prediction that is semantically consistent with the true label, then the prediction for that sample scores a 1 (score_i=1).
If there is no answer in the prediction that is semantically consistent with the true label, then the prediction for that sample scores a 0 (score_i=0).
Your primary task is to provide these scores without delving too deep into the evaluation details.
Required Information for Evaluation:
Prediction: {sentence}
True Label: {ground}
Previous Actions: object_descriptions
Your score for each dimension should be a single number, not a list. Please evaluate the prediction for each sample and return your scores in this format:
1. Logical Consistency: score: []. reason: [].
2. Accuracy: score: []. reason: [].
Your score for each dimension should be a single number, not a list

| Models | Query Type | Temperature |
|--------|-----------|-------------|
| LLaVA | Image | 0.2 |
| MiniGPT-4 | Image | 1.0 |
| mPLUG-Owl | Image | 0.7 |
| InstructBLIP | Video | 1.0 |
| Lynx | Image | 1.0 |
| Otter | Video | 1.0 |

Table 1: Default settings for the models.

## 4  MODEL DEFAULT SETTINGS

## 5  RESULTS OF TEMPERATURE ABLATION EXPERIMENTS

You may include other additional sections here.

| Models | Human Contruction | | | | Automated Generation | | | |
|--------|------|--------|-------|------|------|--------|-------|------|
| | Acc. | Logic. | Spec. | Avg. | Acc. | Logic. | Spec. | Avg. |
| LLaVA-13B | **10.60** | **18.80** | **25.0** | **18.13** | 4.14 | 10.64 | **18.08** | 10.95 |
| LLaVA-7B | 4.00 | 11.60 | 17.20 | 10.93 | 2.14 | 7.68 | 13.10 | 7.64 |
| MiniGPT-4 | 3.80 | 6.40 | 11.20 | 7.13 | 1.08 | 2.20 | 5.07 | 2.78 |
| mPLUG-Owl | 6.40 | 14.20 | 21.60 | 14.07 | 1.52 | 6.10 | 12.88 | 6.83 |
| InstructBLIP-13B | 3.20 | 9.60 | 10.80 | 7.87 | **8.38** | **12.48** | 12.56 | **11.14** |
| InstructBLIP-7B | 1.60 | 10.00 | 9.80 | 7.13 | 0.58 | 6.32 | 6.12 | 4.34 |
| Lynx | 2.80 | 9.60 | 12.80 | 8.40 | 0.34 | 2.78 | 7.76 | 3.63 |
| Otter | 0.80 | 2.20 | 4.20 | 2.4 | 0.14 | 1.36 | 1.98 | 1.16 |

Table 2: Results for the models with a temperature of 0.2.

| Models | Human Contruction | | | | Automated Generation | | | |
|--------|------|--------|-------|------|------|--------|-------|------|
| | Acc. | Logic. | Spec. | Avg. | Acc. | Logic. | Spec. | Avg. |
| LLaVA-13B | **9.40** | 17.80 | **25.20** | **17.47** | 3.62 | 9.56 | **16.8** | **9.99** |
| LLaVA-7B | 4.2 | 14.00 | 17.60 | 11.93 | 2.12 | 7.84 | 12.60 | 7.52 |
| MiniGPT-4 | 3.80 | 8.20 | 15.00 | 9.00 | **3.76** | 4.76 | 8.62 | 5.71 |
| mPLUG-Owl | 8.60 | 16.40 | 21.20 | 15.40 | 1.68 | 6.20 | 13.06 | 6.98 |
| InstructBLIP-13B | 9.02 | **18.52** | 17.87 | 15.14 | 3.06 | **10.04** | 10.86 | 7.99 |
| InstructBLIP-7B | 3.80 | 10.20 | 9.40 | 7.80 | 1.28 | 5.70 | 6.10 | 4.36 |
| Lynx | 1.60 | 8.20 | 14.80 | 8.20 | 0.66 | 3.26 | 7.44 | 3.79 |
| Otter | 0.20 | 1.80 | 3.40 | 1.80 | 0.08 | 1.24 | 1.58 | 0.97 |

Table 3: Results for the models with a temperature of 0.7.

| Models | Human Contruction | | | | Automated Generation | | | |
|---|---|---|---|---|---|---|---|---|
| | Acc. | Logic. | Spec. | Avg. | Acc. | Logic. | Spec. | Avg. |
| LLaVA-13B | **9.20** | **22.60** | 14.40 | **15.40** | **4.38** | 9.02 | **16.76** | **10.05** |
| LLaVA-7B | 6.60 | 11.80 | **17.26** | 12.00 | 1.70 | 6.61 | 10.96 | 6.44 |
| MiniGPT-4 | 3.80 | 6.40 | 11.20 | 7.13 | 2.10 | 2.80 | 7.42 | 4.11 |
| mPLUG-Owl | 6.80 | 16.40 | 21.20 | 14.80 | 1.76 | 6.48 | 12.48 | 6.91 |
| InstructBLIP-13B | 8.80 | 17.60 | 17.20 | 14.53 | 3.14 | **9.96** | 10.56 | 7.89 |
| InstructBLIP-7B | 4.20 | 8.40 | 8.60 | 7.40 | 1.08 | 5.26 | 5.56 | 3.97 |
| Lynx | 2.60 | 10.00 | 13.80 | 8.80 | 0.74 | 3.00 | 6.74 | 3.49 |
| Otter | 0.80 | 2.20 | 3.40 | 2.13 | 0.16 | 1.20 | 1.50 | 0.95 |

Table 4: Results for the models with a temperature of 1.0.