# OpenReview forum: "BENCHMARKING SEQUENTIAL VISUAL INPUT REASONING AND PREDICTION IN MULTIMODAL LARGE LANGUAGE MODELS"
_ICLR.cc/2024/Conference — ICLR 2024 Conference Withdrawn Submission_

### Official Review · Reviewer_aC6M · 2023-10-29

**Soundness:** 2 fair
**Presentation:** 2 fair
**Contribution:** 2 fair
**Rating:** 5
**Confidence:** 5

**Summary:**

This benchmark evaluates the predictive reasoning capabilities of multimodal large language models (MLLMs) in three domains: abstract pattern reasoning, human activity prediction, and physical interaction prediction. The proposed benchmark provides a standardized evaluation framework for MLLMs and can facilitate the development of more advanced models that can reason and predict over complex, long sequences of multimodal input. The paper also presents three evaluation methods to quantify a model's performance in predicting and reasoning about the future based on multi-visual context. This benchmark may reflect the strengths and limitations of current (or future) MLLMs.

**Strengths:**

1. It proposes a novel benchmark for assessing the predictive reasoning capabilities of MLLMs across three important domains, which is a meaningful problem formulation.
2. The paper introduces three evaluation methods that are powered by large language models and can robustly quantify a model's performance in predicting and reasoning the future based on multi-visual context. The quality of the proposed benchmark and evaluation methods is demonstrated through rigorous testing and empirical experiments.
3. The paper provides a standardized evaluation framework for MLLMs, which can facilitate the development of more advanced models that can reason and predict over complex long sequences of multimodal input.

**Weaknesses:**

1. While the paper offers an assessment of MLLMs' performance across benchmarks (encompassing abstract pattern reasoning, human activity prediction, and physical interaction prediction), it does not delve deeply into the specific types of errors these models tend to make. A more granular insight into the exact nature of the mistakes that MLLMs are prone to would have been invaluable.

2. The study could benefit from a comprehensive examination of the impact of MLLMs' scale, particularly concerning their emergent ability to predict forthcoming events based on contextual cues. A more detailed exploration of the kinds of errors that MLLMs are vulnerable to, and how these inaccuracies might be reduced with the model's scaling (ranging from 7B to 13B models like Vicuna and InstructBLIP, and further expanding to ~1700B as in GPT4-Vision), would augment the depth of the analysis.

3. The paper lacks of discourse on the potential advantages of pretraining MLLMs' vision modules on video datasets. Such pretraining could equip MLLMs with a better grasp of world dynamics, thereby enhancing their predictive capabilities in anticipating upcoming events based on context. This could be especially potent when contrasted with MLLMs whose vision modules have only been pretrained on static image datasets.

**Questions:**

Please see weaknesses.

**Details Of Ethics Concerns:**

N.A.

---

### Official Review · Reviewer_v1k7 · 2023-10-31

**Soundness:** 2 fair
**Presentation:** 2 fair
**Contribution:** 2 fair
**Rating:** 3
**Confidence:** 4

**Summary:**

The authors propose a benchmark to assess the predictive reasoning ability of the current large language models. Specifically, they look at abstract pattern reasoning, human activity prediction, and physical interaction prediction tasks. They find that there is a substantial gap between the current models and upper bounds on these datasets. Interestingly, they find that the image models outperforms the video models for many of the tasks too. Despite being a good problem to address, the paper is not well-written and motivated which makes it difficult to buy in its current state.


Comments:

- I agree that predictive reasoning is an important skill for AI systems to have. However, why should one expect multimodal LLMs which are not explicitly trained for predictive reasoning to be good at these? The current introduction does not convince me how and why people would want to use single-image MLLMs for multiple image sequence tasks.
- In Section 2.2.1, the paper makes an incorrect claim that there has been no prior work on evaluating MLLM’s ability to predict over sequential visual input. I point that [1,2] include multiple-images in their work. Even if we ignore these works, there have been several datasets including Robust Change Captioning, NLVR2, ImageCoDE, .. IconQA (Table 5 of [1]) which does multi-image evaluation of AI models. The paper ignores all this prior work and overestimates its novelty. I clarify that these datasets are different from video input based works mentioned in Section 2.2. I fail to understand why we should care about the new dataset proposed in the paper given the existing multi-image datasets.
- How is Abstract Reasoning Patterns task different from IconQa examples? https://iconqa.github.io/explore.html
- Action recognition is a classic video task and there are existing datasets for it such as Kinetics-400.
- Section 3.2 is poorly written which makes it hard to understand how the model evaluation is done. Specifically, what is the choice of g? Most of the equations 1-6 seem unnecessarily complicated. I think they can be compressed to be more coherent.
- The paper proposes many new metrics but lacks grounding with human evaluation. Do the humans score the models on the Specificity as their method does? How noisy is the metric?
- Given LLaVA performs the best, I would have liked to see any analysis on the LlaVA instruction dataset. How many instructions in its dataset enforce approximate predictive reasoning?

References:

[1] https://arxiv.org/pdf/2308.06595.pdf
[2] https://arxiv.org/pdf/2304.14178.pdf

**Strengths:**

Mentioned in the comments

**Weaknesses:**

Mentioned in the comments

**Questions:**

Mentioned in the comments

---

### Official Review · Reviewer_wwba · 2023-10-31

**Soundness:** 2 fair
**Presentation:** 2 fair
**Contribution:** 3 good
**Rating:** 3
**Confidence:** 4

**Summary:**

This paper introduces three tasks for evaluating multimodal large language models on predictive reasoning in the areas of abstract pattern reasoning, human activity prediction, and physical interaction prediction. The authors construct new datasets using the ground truth from the original datasets and proposes GPT4-based evaluators based on a few aspects including accuracy, specificity, plausibility, and diversity. The experiment results demonstrates the deficiency of current models on abstract pattern reasoning and physical interaction prediction while have reasonable performance on the human activity prediction task. The effectiveness of evaluators are also verified by proposed experiments.

**Strengths:**

1. The benchmark itself is new and could work as a testbed for future visual language models.

**Weaknesses:**

1. More examples of the dataset should be provided in order to demonstrate effectiveness.
2. Notations used in Sec 3.2 are complex and a bit misleading.
3. Not much details of the evaluator are given although they leads to the conclusion of the paper.
4. The models used in the paper are not designed for multiple images, so there should be more discussion on whether the poor performance is due to the unseen prompt format.

Minor issues:
1. Sec 3, physucal -> physical
2. please use \citep instead of \citet
3. In supplementary, quary -> query

**Questions:**

1. How is the evaluators used in the papers constructed? The scale used in supplementary is not in match with the results in paper.
2. Is there any particular reason using instead of existing datasets, e.g. Raven's Progressive Matrices [1]. How is the additional dataset being generated?
3. Why is there a training set for MT-ACT? Is any model used in the paper being trained on the datasets? If so, please provide the details.
4. How is the Acc in Table 6 being calculated?

[1] https://arxiv.org/abs/1903.02741

---

### Official Review · Reviewer_SwvU · 2023-10-31

**Soundness:** 3 good
**Presentation:** 3 good
**Contribution:** 2 fair
**Rating:** 5
**Confidence:** 4

**Summary:**

This paper introduces a benchmark that assesses the predictive reasoning capabilities of MLLMs across diverse scenarios. The benchmark targets three domains: abstract pattern reasoning, human activity prediction, and physical interaction prediction. The paper evaluates current state of the art LLMs on the benchmark.

**Strengths:**

• The paper addresses an important problem.
• The paper addresses each component task and dataset in detail.
• The paper includes state of the art multi-modal LLMs such as LLaVA and InstructBLIP.

**Weaknesses:**

• Comparison to existing benchmarks for multi-modal LLMs is missing: “Perception Test: A Diagnostic Benchmark for Multimodal Video Models, NeurIPS 2023” already proposes a benchmark suite which includes temporal sequence prediction tasks such as tracking and questions on human actions. “SEED-Bench: Benchmarking Multimodal LLMs with Generative Comprehension, arXiv 2023” contains questions on action recognition, action prediction and procedure understanding.

• There are already many existing datasets for evaluation of each of the component tasks: abstract pattern reasoning tasks: RPM prediction “Raven: A dataset for relational and analogical visual reasoning, CVPR 2019”, human-centric activity task: ActivityNet-QA “ActivityNet-QA: A Dataset for Understanding Complex Web Videos via Question Answering, AAAI 2019”, “STAR: A Benchmark for Situated Reasoning in Real-World Videos, NeurIPS 2021”. It is unclear why the proposed data splits are better than existing benchmarks.

• It is unclear from the paper, the difficulty level of each task. For the human-centric activity task  task, the paper chooses 309 and 260 video segments from ActivityNet and Charades respectively. It is unclear how challenging these scenarios are. It would be helpful to include non-LLM based supervised baselines to calibrate the difficult of each task. The paper should include more qualitative examples to highlight the difficulty level of each task.

• Eqs 1-6 seem more like decorative math and are hard to parse. Their realizations in page 6 are much easier to understand and are slight variations of existing evaluation protocols.

• It is unclear how Plausibility, Diversity and Specificity are computed exactly.

• For the Multiple Gold Answer Evaluator, it is unclear how exactly the point-based scoring system is implemented.

• For evaluation of ActivityNet captions standard  metrics such as BLEU and Rouge should also be used.

• The benchmark could also integrate an “overall” metric for a global ranking across all tasks.

• The paper could also include GPT-4V as it is the current state-of-the-art multi-modal LLM.

**Questions:**

• The paper should include a more through comparison to prior multi-modal LLM benchmarks.
• The paper should explain in more detail why each component sub-task was chosen.
• Many of the evaluation metrics, e.g., Plausibility, Diversity and Specificity, are not described in detail.